# Rapid and scalable photocatalytic C(sp²)–C(sp³) Suzuki–Miyaura cross-coupling of aryl bromides with alkyl boranes

Ting Wan[1,2,6], Luca Capaldo [1,3,6], Jonas Djossou[1], Angela Staffa[1,4], Felix J. de Zwart [5], Bas de Bruin [5] & Timothy Noël [1] ✉

In recent years, there has been a growing demand for drug design approaches that incorporate a higher number of sp³-hybridized carbons, necessitating the development of innovative cross-coupling strategies to reliably introduce aliphatic fragments. Here, we present a powerful approach for the light-mediated B-alkyl Suzuki–Miyaura cross-coupling between alkyl boranes and aryl bromides. Alkyl boranes were easily generated via hydroboration from readily available alkenes, exhibiting excellent regioselectivity and enabling the selective transfer of a diverse range of primary alkyl fragments onto the arene ring under photocatalytic conditions. This methodology eliminates the need for expensive catalytic systems and sensitive organometallic compounds, operating efficiently at room temperature within just 30 min. We further demonstrate the translation of the present protocol to continuous-flow conditions, enhancing scalability, safety, and overall efficiency of the method. This versatile approach offers significant potential for accelerating drug discovery efforts by enabling the introduction of complex aliphatic fragments in a straightforward and reliable manner.

The advancement of transition metal-catalyzed cross-coupling reactions has revolutionized the field of pharmaceutical and agrochemical synthesis, significantly enhancing the efficiency of accessing potent molecules. While these transformations have exhibited exceptional fidelity, their widespread success has inadvertently fostered a reliance on generating structurally analogous, planar compounds, thereby introducing a bias into the realm of small molecule design. To counteract this trend, the concept of "diversity-oriented synthesis" was introduced with the primary goal of facilitating the synthesis of more intricate molecules[1]. This paradigm shift issued a strong call to practitioners of synthesis, prompting a rapid and innovative response, as they embarked on the creation of

an extensive array of sophisticated methodologies geared towards the incorporation of sp³-fragments[2].

The B-alkyl Suzuki–Miyaura cross-coupling (SMC) reaction (Fig. 1A) stands out as a robust methodology in this field, involving the coupling of alkyl boranes with aryl or vinyl (pseudo)halides[3,4]. Compared to other commonly employed cross-coupling reactions, such as Negishi[5] or Kumada[6,7] reactions, the B-alkyl SMC offers several distinct advantages. First, the alkyl borane coupling partner can be generated under exceptionally mild conditions via hydroboration of olefins. This approach not only tolerates a wide range of functional groups but also eliminates the need for unstable alkylzinc or Grignard reagents[8]. Second, the boron-derived by-products produced during the reaction

[1]Flow Chemistry Group, van't Hoff Institute for Molecular Sciences (HIMS), University of Amsterdam, 1098 XH Amsterdam, The Netherlands. [2]The Research Center of Chiral Drugs, Innovation Research Institute of Traditional Chinese Medicine, Shanghai University of Traditional Chinese Medicine, Shanghai 201203, China. [3]SynCat Lab, Department of Chemistry, Life Sciences and Environmental Sustainability, University of Parma, 43124 Parma, Italy. [4]Merck Healthcare KGaA, Frankfurter Str. 250, 64293 Darmstadt, Germany. [5]Homogeneous, Supramolecular and Bioinspired Catalysis Group (HomKat), van't Hoff Institute for Molecular Sciences (HIMS), Universiteit van Amsterdam (UvA), 1098 XH Amsterdam, The Netherlands. [6]These authors contributed equally: Ting Wan, Luca Capaldo. ✉e-mail: t.noel@uva.nl

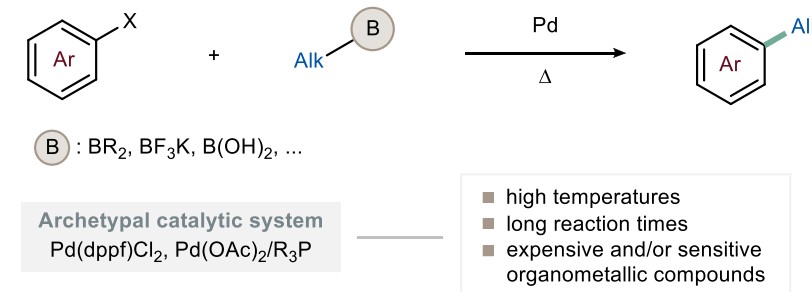

**A  B-alkyl Suzuki-Miyaura cross-coupling**

**B  Light-mediated B-alkyl Suzuki-Miyaura cross-coupling**

**C  This work**

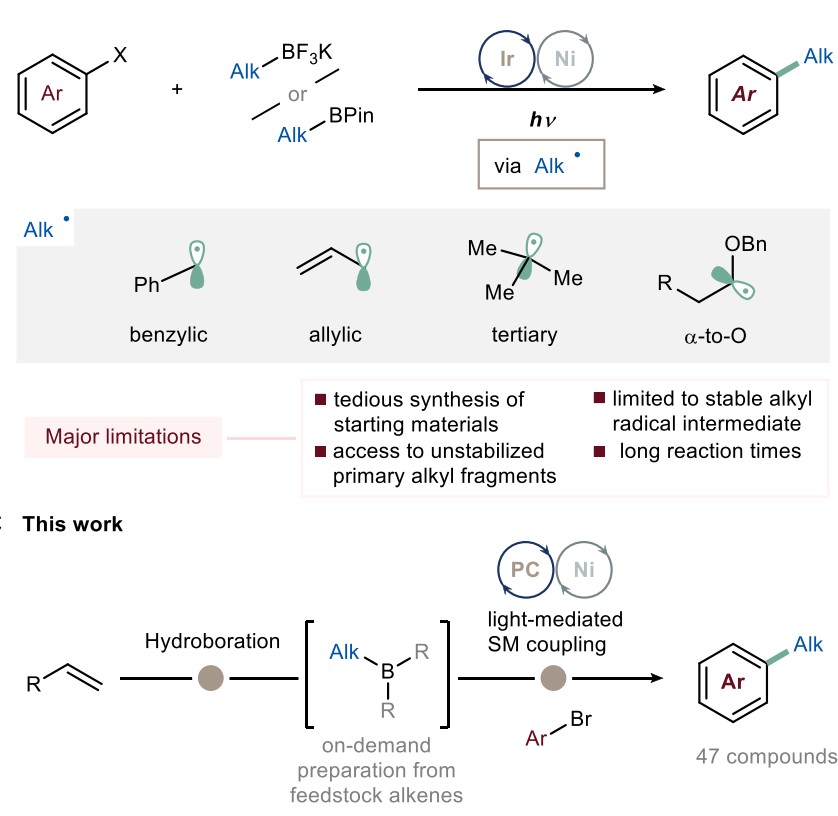

**Fig. 1 | Context, state-of-the-art and this work. A** The B-alkyl Suzuki−Miyaura cross-coupling is a powerful approach for the formation of C(sp³)−C(sp²) bonds: the archetypal Pd-based catalysis typically requires refluxing solvents and relatively long reaction times. **B** Light-mediated B-alkyl Suzuki−Miyaura cross-coupling reaction via single-electron transmetalation: this approach is mainly restricted to the generation of stabilized radicals. **C** This work: photocatalyzed B-alkyl Suzuki −Miyaura cross-coupling with alkyl borane to append primary fragments onto arenes in short reaction times. dppf: 1,1 ′-Bis (diphenylphosphino)ferrocene. Pin: pinacol. Alk: alkyl group. Ar: aryl group. PC: photocatalyst.

pose manageable toxicity concerns[9]. Third, the reaction conveniently proceeds under non-dry conditions[4]. While state-of-the-art catalysts for coupling alkyl boranes with aryl halides are primarily palladium-based (e.g., Pd(dppf)Cl₂[10–12] and Pd(OAc)₂[13,14])[3], these reactions often require refluxing solvents, elevated temperatures, and extended reaction times, all performed under strictly inert conditions. The above can significantly hinder high throughput experimentation for library generation in routine medicinal or process chemistry applications[15].

In recent years, the application of light-mediated synthesis, which harnesses photonic energy to initiate chemical reactions, has gained substantial traction within both academic and industrial organic synthesis domains[16]. Within the context of C(sp³)−C(sp²) bond

formation, metallaphotoredox chemistry has emerged as a corner-stone methodology[17]. Particularly noteworthy are the elegant techniques that leverage light-mediated approaches for B-alkyl SMC, primarily relying on single-electron transmetalation[18–20]. In these methods, a photocatalyst is employed to generate carbon-centered radicals from boron derivatives (such as trifluoroborate potassium salts and boronic esters). These intermediates are subsequently captured by a metal complex, typically nickel, resulting in the formation of the desired C(sp³)−C(sp²) bond (Fig. 1B). While these approaches have found widespread adoption for the functionalization of benzylic, allylic, α-to-O positions[19,21–23], they are primarily hampered by their inability to efficiently generate unstabilized, primary carbon-centered

radicals. Additionally, some of these methods require extended irradiation times[22], which can hinder their practical utility and scalability. It is important to note that, albeit commercially available trifluoroborate salts and simple boronic acid ester derivatives exist, complex molecule derivatives are not readily accessible due to the use of harsh reaction conditions or metal catalysis required for their synthesis, which may not always be compatible with the diverse functionalities present in complex organic structures.

Recognizing the burgeoning interest within the synthetic chemistry community, we set out to develop a unified, practical, and mild light-mediated B-alkyl SMC methodology to address the aforementioned limitations, thereby introducing a new dimension to cross-coupling reactions. At the outset of our investigations, we identified alkyl boranes as particularly intriguing candidates for alkyl coupling partners. This choice stemmed from their facile generation through the regioselective hydroboration of readily available alkenes, with hydroboration consistently following anti-Markovnikov regiochemistry[24]. This undertaking commenced with a remarkable degree of uncertainty, given that the suitability of alkyl boranes as precursors for alkyl fragments under photocatalytic conditions had not been previously established, despite their widespread utilization in thermal chemistry[3,11,12,25–30].

In this study, we present the convenient utilization of said alkyl boranes as versatile coupling partners, establishing a robust and scalable platform for constructing C(sp$^3$)−C(sp$^2$) bonds via Suzuki−Miyaura cross-coupling (SMC), made possible by integrating photocatalysis and nickel catalysis (Fig. 1C). Remarkably, nickel offers several distinct advantages over palladium, including cost-efficiency and, of particular relevance to our objective, a reduced propensity for β-hydride elimination when working with alkyl fragments[31]. Our protocol specifically caters to the introduction of non-activated alkyl fragments, making it complementary to other metallaphotoredox strategies. The effectiveness of this coupling strategy is exemplified by successfully incorporating complex alkyl fragments derived from natural products. Additionally, we demonstrate the potential of flow technology in realizing a scalable, safe and streamlined process, combining both the hydroboration and the light-mediated B-alkyl SMC reaction[32].

## Results and discussion
### Method optimization
We initiated our research by investigating the cross-coupling of commercially available triethyl borane 1a with methyl 4-bromobenzoate 2a to yield alkyl arene 3. Through a preliminary screening of various photocatalysts (Table 1, Entries 1−3) under blue light irradiation ($\lambda = 456$ nm), we discovered that the organic photocatalyst 4CzIPN outperformed the benchmark Ir-based photocatalyst used in previous reports on light-mediated B-alkyl SMC reactions[18–20,33]. In contrast, the highly oxidizing acridinium photocatalyst did not prove as effective (Table 1, Entry 3). Notably, even under non-dry conditions (Table 1, Entry 4), we obtained product 3 with only a slight reduction in yield, which demonstrates the robustness of the protocol. Diminished yields were observed when using a sub-stoichiometric amount of the base or lower amounts of the alkyl borane (Table 1), Entries 5 and 6, underscoring the significance of the base for efficient catalysis and confirming that only one of the alkyl groups transfers to nickel. Changing the nickel source minimally impacted the reaction, as nickel bromide delivered nearly identical yields (Table 1, Entry 7). Encouragingly, the reaction was found to be complete after 30 min of irradiation (Table 1, Entry 8). Extended irradiation times did not result in any detectable product decomposition, even after 3 h of irradiation. Therefore, for the sake of generality, we established 3 h as the ideal reaction time to carry out the reaction scope. Notably, the transformation proceeded smoothly even with reduced loading of both the photocatalyst and the nickel catalyst (Table 1, Entry 9). The aforementioned results highlight the robustness of this B-alkyl SMC methodology, which is particularly

useful to facilitate process chemistry applications[34]. Next, several control experiments were conducted, demonstrating that the absence of either the photocatalyst or the nickel catalytic system resulted in no product formation (Table 1, Entries 10 and 11). Furthermore, omission of the base yielded only traces of 3 (Table 1, Entry 12); a similar result was obtained when performing the reaction at elevated temperatures in the dark (Table 1, Entry 13). Additional details and optimization experiments can be found in the Supplementary Information.

### Reaction scope
Having established optimal reaction conditions (Table 1, Entry 8), our next objective was to assess the scope of the light-mediated B-alkyl SMC reaction (Fig. 2). As expected, aryl bromides with electron-withdrawing groups displayed favorable performance, resulting in good isolated yields for compounds 3–8 (40–78%). Regarding the substitution pattern on the aromatic ring (3−5), we observed that the presence of an ortho substituent led to lower yields. Subsequently, we shifted our attention to electron-rich aryl bromides, known for their reluctance to undergo oxidative addition by the nickel catalyst. Although challenging, we obtained satisfactory yields for compounds 9 and 10 (40–42%). The cross-coupling reaction also tolerated a BPin functional group (compound 11, 65%), providing opportunities for further diversification. Notably, this particular substrate would have posed inherent difficulties for previously developed approaches (Fig. 1B)[18,19]. When subjected to the optimized reaction conditions, 4-bromo-2-chloro-1-fluorobenzene underwent functionalization solely at the C−Br bond, yielding compound 12 in excellent yield (88%). Recognizing the significance of heterocycles in pharmaceutical compound synthesis, we applied our reaction conditions to a wide range of heteroaryl bromides, including pyridines, quinolines, indoles, indazoles, and pyrazolopyridines. All of these substrates yielded the expected products in good to excellent yields (13–25, 34–87%).

Next, our focus shifted to the alkyl borane coupling partner, which can be conveniently synthesized on demand through the hydroboration of readily available or easily prepared alkenes with BH$_3$ (known as Brown hydroboration)[35]. Upon subjecting alkyl boranes derived from 1,1-disubstituted olefins to the optimized conditions (referred to as Method A in Fig. 3), the desired products were obtained with excellent yields and exhibited high anti-Markovnikov regioselectivity (54–77%, >20:1 regioisomeric ratio, 26-27). Remarkably, an allyl chloride could also participate in the transformation, resulting in a 47% yield of product 28. Alkyl boranes derived from cycloalkenes underwent smooth arylation, affording products 29 and 30 in very good yields (67–83%), thus demonstrating the compatibility of secondary carbons with the transformation. Notably, even an estradiol derivative served as a suitable substrate under these conditions, leading to the isolation of compound 31 with a 43% yield and exquisite regioselectivity.

As anticipated, regioselectivity issues arose when sterically unhindered alkenes with long aliphatic chains were subjected to the conditions of Method A. For instance, ether 32 could be obtained in a good yield, albeit as a mixture of regioisomers (74%, rr 10:1). To address this challenge, we capitalized on existing knowledge of hydroboration methods[36], and introduced 9-BBN as the hydroboration agent (referred to as Method B, see also Section 2.3 in the Supplementary Information). Armed with this new set of conditions, we successfully achieved excellent regioselectivity in the synthesis of compound 32 (59%, rr > 20:1). Similarly, a series of long-chain alkenes incorporating multiple functional groups proved amenable to these conditions, yielding the expected products as single regioisomers with excellent yields (33-34, 72–86%). The presence of an acetal functionality was also tolerated, leading to the formation of product 35 in a 30% yield. Moreover, the transformation worked well with vinylcyclohexene, affording compound 36 in an excellent yield of 90%. Intriguingly, when phenyl-containing alkenes were

**Table 1 |** [a] **Yields determined by ¹H-NMR, CH₂Br₂ as external standard.** [b] **74% recovery of starting material.** [c] **63% recovery of starting material**

| Entry | Variation from conditions | Yield (3)[a] |
|---|---|---|
| 1 | None | 93 |
| 2 | (Ir[dF(CF₃)ppy]₂(dtbbpy))PF₆ (5 mol%) as the photocatalyst | 82 |
| 3 | Mes-AcrClO₄ (5 mol%) as the photocatalyst | 23 |
| 4 | In the presence of H₂O (10 equiv.) | 85 |
| 5 | 2,6-lutidine (0.2 equiv.) | 21 |
| 6 | Et₃B (0.33 equiv.) | 33 |
| 7 | NiBr₂·glyme (10 mol%) | 90 |
| 8 | **Reaction time: 3 h** | **92** |
| 9 | 4CzIPN (2 mol%), NiCl₂·glyme (5 mol%), dtbbpy (5 mol%), 3 h | 82 |
| 10 | No photocatalyst | n.d. |
| 11 | No NiCl₂·glyme nor dtbbpy | n.d. |
| 12 | No 2,6-lutidine | 8[b] |
| 13 | No light, T: 80 °C | 7[c] |

4CzIPN: 2,4,5,6-tetrakis(9H-carbazol-9-yl) isophthalonitrile. Mes-AcrClO₄: 9-Mesityl-10-methylacridinium perchlorate. Dtbbpy: 4,4'-di-*tert*-butyl-2,2'-dipyridyl. PC: photocatalyst. n.d.: not detected. Optimized condition in bold.

employed as substrates, the formation of alkylarenes **37** and **38** was observed (82-85%), while significant migration of the nickel center to the benzylic position was not detected[37]. Once again, this underscores the potential of our methodology for functionalizing alkyl chain termini.

Our focus then shifted towards the functionalization of biologically and medicinally relevant scaffolds. Interestingly, protected piperidine was well tolerated within the protocol. Specifically, when 4-vinyl piperidine was subjected to the optimized reaction conditions (Method B), compound **39** was obtained in a 79% yield after isolation. While unprotected amines proved unsuccessful, likely due to detrimental N–B interactions, our reaction conditions successfully engaged tertiary amines and protected primary amines (**40-42**). Five-membered heterocycles delivered the expected products **42** and **43** in 60% and 56% yield, respectively.

The functionalization of β-pinene proceeded smoothly, providing a single diastereomer in excellent yield (**44**, 90%). The hydroarylation of (−)-limonene resulted in a mixture of diastereomers (**45**, 84%, dr 1:1). However, the regioselectivity exhibited was remarkable, functionalizing only the exocyclic olefin. A similar trend was observed for the citral-derived alkene, wherein selective functionalization of the primary olefin position occurred (**46**, 40%), leaving the more substituted olefins untouched. Finally, a sugar and a glucocorticoid derivative underwent efficient functionalization, providing the targeted products **47** (36%) and **48** (50%, dr 10:1).

As a final observation, we found that utilizing the pinacol ester of allyl boronic acid as the substrate in the hydroboration-cross-coupling process resulted in a 47% yield for the isolated product **49**. This outcome underscores the tolerance of our method towards BPin functional groups. Product **49** is highly compatible with the amino radical transfer strategy recently reported by scientists at Sanofi[18]. Interestingly, when subjected to the conditions outlined in their study, we successfully obtained product **50** with a yield of 63%. This consecutive transformation highlights the orthogonal nature of these two approaches.

## Mechanistic investigation

In order to gain insights into the mechanistic aspects of the developed photocatalytic B-alkyl SMC, a series of experiments were conducted. First, we investigated whether alkyl boranes formed "ate" complexes with 2,6-lutidine. However, it is worth noting that 2,6-lutidine is considered a weak Lewis base, which raised doubts about direct complexation of the borane by the heterocycle[33]. To investigate this further, titration experiments were performed and followed both via ¹H-NMR and ¹¹B-NMR. No apparent NMR signal shifts were observed when mixing 2,6-lutidine and triethyl borane (1:1), providing initial evidence against complexation (Fig. 4A). Additionally, when the organic base was replaced with inorganic bases, such as K₃PO₄ and Cs₂CO₃, similar yields were obtained (85% and 83%, respectively), thus indicating the absence of any specific interaction between 2,6-lutidine and the borane (Fig. 4A)[18].

Next, our efforts were directed toward detecting open-shell intermediates within our reaction conditions. To this end, we employed Electron Paramagnetic Resonance (EPR) spectroscopy at low temperature (10 K). Inside a J-Young EPR tube, the catalytic mixture was irradiated with 456 nm light for 30 s and then rapidly cooled in liquid nitrogen while still under irradiation, allowing for the acquisition of the spectrum presented in Fig. 4B (black trace). Analysis of the spectrum revealed two doublet components, corresponding to an isotropic organic radical with $g_{iso} = 2.002$, derived from the photocatalyst, and a nearly axial metal-centered radical with $g_{xyz} = [2.265\ 2.23\ 2.047]$[38]. The g-values of the axial signal correspond to those documented for a paramagnetic nickel species (Ni^III or Ni^I). This particular species is probably the resting state in the nickel catalytic cycle. It could be a Ni^III species, but given the instability of these species and the need for specific ligands for their stabilization, which is not the case in our scenario, we suggest that a Ni^I species is accountable for this signal[39–43]. Despite several attempts, we were unable to detect or trap ethyl radicals derived from **1a** (see Supplementary Discussion).

Next, several representative reactions were conducted in the presence of an excess of butylated hydroxytoluene (BHT), a

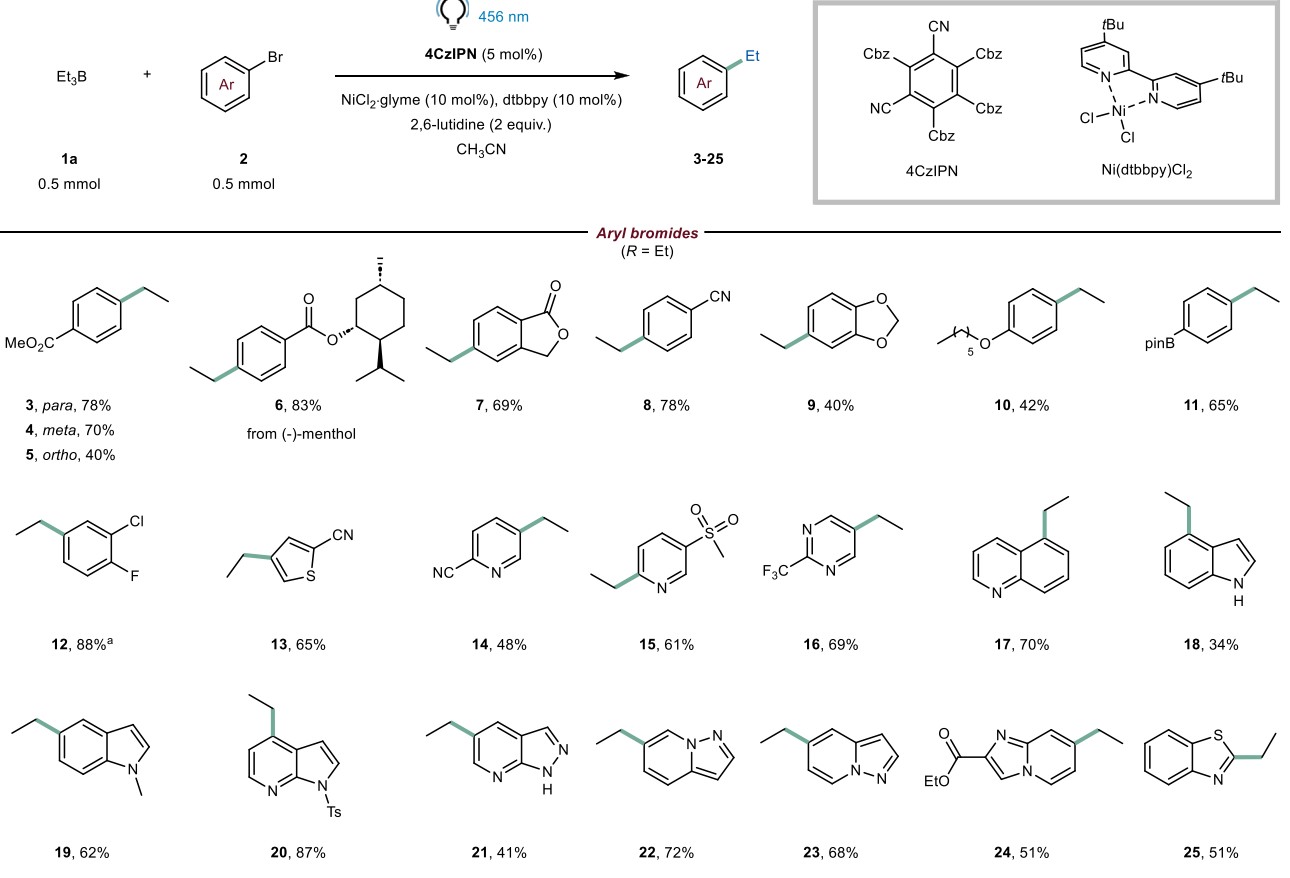

**Fig. 2 | Survey of the aryl bromides.** Conditions: 4CzIPN (5 mol%), NiCl$_2$-glyme (10 mol%), dtbbpy (10 mol%), 2,6-lutidine (2.0 equiv.), **1a** (1.0 equiv.) and **2** (0.5 mmol) in CH$_3$CN (0.1 M). The solution was N$_2$-bubbled and subsequently irradiated with a blue LED ($\lambda = 456$ nm) for 3 h (see Section 1.5 in Supplementary Information). Yields are meant after isolation, unless otherwise stated. [a] NMR yield is given because of the volatility of the compound.

well-known radical scavenger. The presence of BHT did not impede the cross-coupling reaction, and the expected products were obtained with unaltered yields (see Section 2.1.5 in Supplementary Information). In contrast, the addition of 1 and 2 equivalents of TEMPO effectively quenched the reaction, yielding the corresponding TEMPO-Et adduct in 35% (along with 33% of **3**) and 63% $^1$H NMR yield, respectively. Notably, when light was omitted, no TEMPO-Et adduct was detected via GCMS or NMR[44]. These results can be rationalized by the rapid interception of the putative alkyl radical by TEMPO, as opposed to BHT. It is worth noting that BHT has been previously reported to intercept even sterically hindered primary radicals[45].

To elucidate the underlying reasons for these seemingly contradictory results, we conducted irradiation experiments using an O$_2$-free $5 \cdot 10^{-3}$ M CH$_3$CN solution of the photocatalyst in the presence of Et$_3$B (1 equiv.) while excluding 2,6-lutidine. In this setting, we observed the decyanative ethylation of 4CzIPN, an observation that was entirely inhibited in the presence of TEMPO. Interestingly, a comparable photochemical transformation of the photocatalyst core had been previously reported by König and co-workers[46] when using phenylacetic acids as radical precursors. In their work, the authors noted that alkyl radicals generated through the reductive quenching of 4CzIPN had the capacity to add onto the photocatalyst core, leading to its alkylation. In light of these findings, we posit that 4CzIPN is also capable of producing alkyl radicals from alkyl boranes. Notably, under optimized conditions, the photocatalyst can be fully recovered after the reaction, suggesting the absence of this decomposition pathway. This is likely due to the highly effective trapping of radicals by the nickel complex en route to the formation of product **3**. On the contrary, in the absence of the base, the photocatalyst undergoes decomposition, highlighting

the pivotal role played by the base in preserving the photocatalyst. Further investigations on this aspect are currently underway in our laboratories.

The determination of the quantum yield of the process via ferrioxalate actinometry[47,48] revealed a value of 0.27, indicating that a self-sustained Ni$^{I/III}$ cycle[47] is unlikely to be the primary scenario for this B-alkyl SMC. Moreover, we observed that the reaction did not proceed in the absence of light, even after a short initial irradiation period of 3 min.

To gain further insights into the dual catalytic system, we conducted additional experiments (Fig. 4C). Remarkably, in a stoichiometric experiment, where 1 equivalent of the oxidative addition complex Ar–Ni$^{II}$–Br was stirred with **1a** in the dark without a photocatalyst for 3 h, a 42% $^1$H-NMR yield of compound **3** was obtained. This result is consistent with a scenario in which polar transmetalation and reductive elimination proceed smoothly, albeit at a slower rate compared to ideal conditions (30 min, as depicted in the kinetic profile of the reaction in Supplementary Fig. 14). In contrast, when the reaction was conducted without light and photocatalyst, using 10 mol% of the oxidative addition complex Ar–Ni$^{II}$–Br as the active metal species, only traces of the product were observed. This suggests that oxidative addition might represent the turnover-limiting step within the nickel cycle. In fact, when the reaction was conducted by replacing the photocatalyst/Ni$^{II}$ system with Ni(COD)$_2$ (10 mol%) under strictly inert conditions in the dark, only traces of product **3** were detected. Together with the experiment displayed in Fig. 4Ci, this shows that oxidative addition under thermal conditions is sluggish, lending support our hypothesis. In stark contrast, when the Ni(COD)$_2$ experiment was performed in the presence of photocatalyst and light, reactivity was

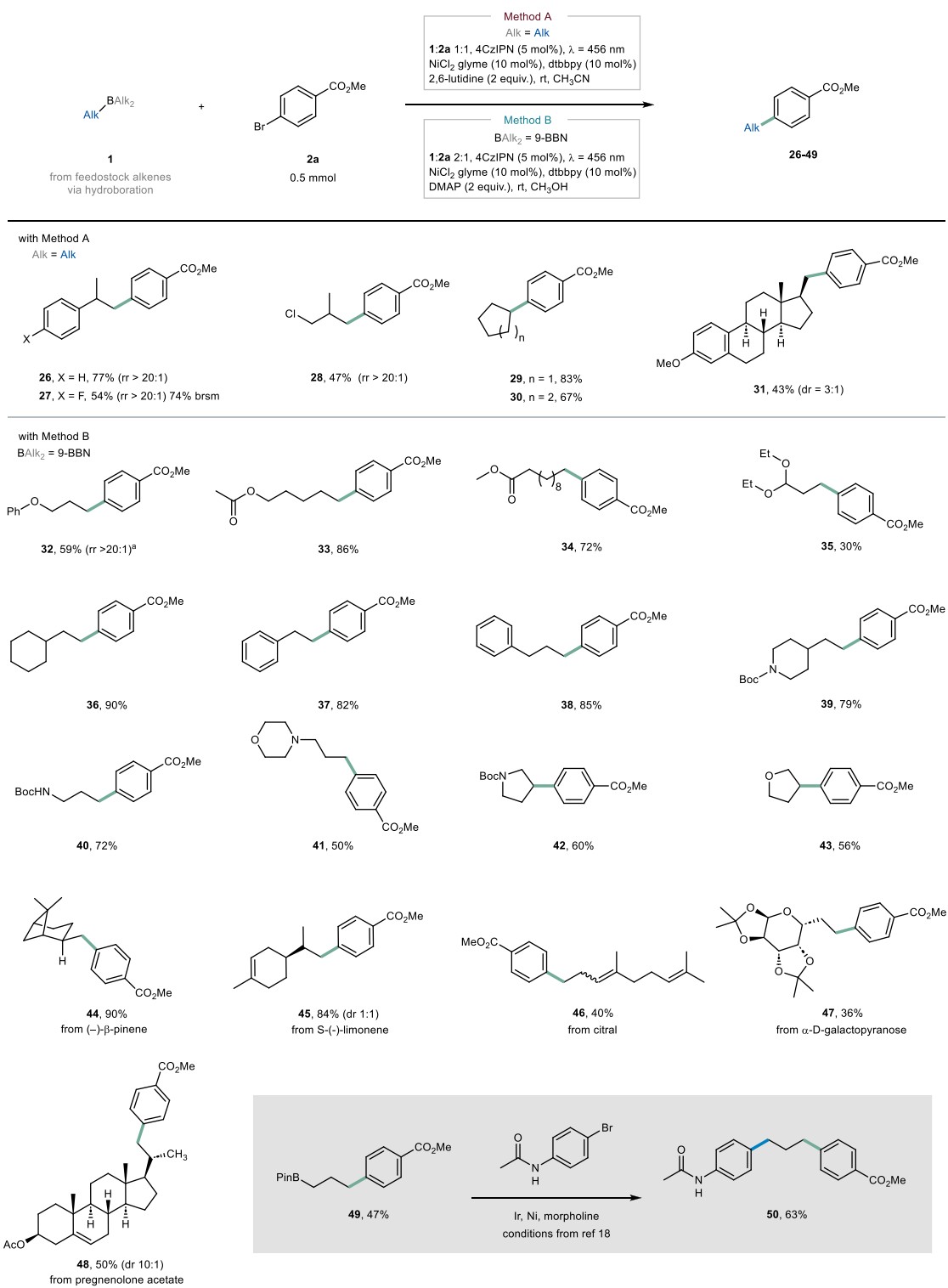

**Fig. 3 | Survey of the alkyl boranes.** Conditions for Method A: 4CzIPN (5 mol %), NiCl$_2$·glyme (10 mol %), dtbbpy (10 mol %), 2,6-lutidine (2 equiv.), **1** (1.0 equiv.) and **2** (0.5 mmol) in CH$_3$CN (0.1 M). The solution was N$_2$-bubbled and irradiated with a blue LED ($\lambda$ = 456 nm) for 3 h. Conditions for Method B: 4CzIPN (5 mol %), NiCl$_2$·glyme (10 mol %), dtbbpy (10 mol %), 4–dimethylaminopyridine (2 equiv.), **1** (2.0 equiv.) and **2** (0.5 mmol) in CH$_3$OH (0.1 M). The solution was N$_2$-bubbled and irradiated with a blue LED ($\lambda$ = 456 nm) for 3 h. Alkyl boranes were generated on-demand from the corresponding alkenes via hydroboration with BH$_3$ (for Method A) or 9-BBN (for Method B). Yields reported are after isolation. [a] with Method A: 74%, rr 10:1. See Section 2.3 in the Supplementary Information for additional details. rr: regioisomeric ratio. dr: diastereomeric ratio. Brsm: based on recovered starting material. 9-BBN: 9-borabicyclo[3.3.1]nonane.

restored, resulting in a 77% $^1$H-NMR yield of product **3**. In light of the above, we propose that radical addition onto a Ni$^0$ species is likely responsible for the boost in the oxidative addition step and, thus, the enhanced reactivity[49].

On a different note, when substrate **S2** was subjected to our optimized conditions, we detected traces of a product resulting from chain-walking via iterative β-hydride elimination and migratory insertion (compound **38γ**). Notably, when ligand **L2**, known to promote

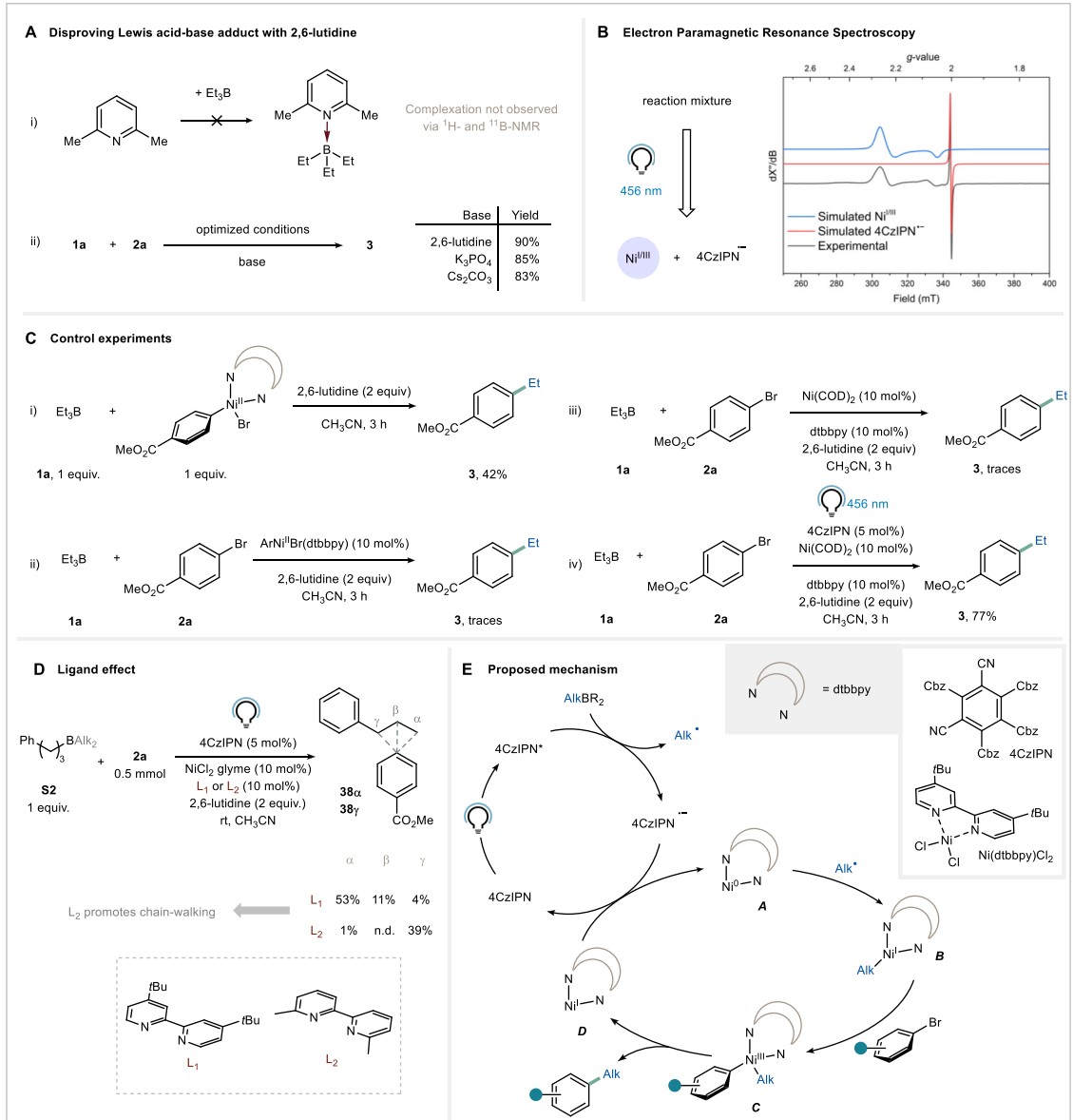

**Fig. 4 | Mechanistic investigation. A** Results of NMR complexation studies disproving Lewis acid-base adduct with 2,6-lutidine. **B** Electronic-Paramagnetic Resonance (EPR) results. CW X-Band EPR spectrum of catalytic mixture in 5:1 PrCN:MeCN at 10 K, after 30 s irradiation with blue light (Mw Freq = 9.6503 GHz). Experimental spectrum (black), simulated spectrum of a paramagnetic Ni species (blue, $S = 1/2$, $g = [2.265\ 2.23\ 2.047]$), simulated spectrum 4CzIPN radical anion (red, $S = 1/2$, $g_{iso} = 2.002$). L$_1$: 4,4'-di-*tert*-butyl-2,2'-dipyridyl; L$_2$: 6,6'-dimethyl-2,2'-dipyridyl. **C** Control experiments. **D** Effect of the ligand on the reaction outcome. **E** Proposed mechanism.

chain-walking[50,51], was employed, selectivity was completely shifted towards the benzylic position (Fig. 4D). It is worth mentioning that while the elementary processes of $\beta$-hydride elimination and migratory insertion are frequently associated with a Ni$^{II}$ species[51,52]; instances of chain-walking have also been documented for both Ni$^{I}$ and Ni$^{III}$ species[53].

To further explore the reaction kinetics, we conducted an investigation into the dependence of the initial rate on various factors, including substrate concentration, nickel catalyst concentration, and base concentration in the reaction between **1a** and **2a**. The progress of the reaction and the formation of product **3** were monitored using $^1$H-NMR. This study revealed that, under the optimized conditions, the reaction operates in a photon-limited regime, indicating that the rate is primarily determined by the availability of photons. Intriguingly, the reaction exhibited zeroth order kinetics with respect to all other reaction components. A related

Hammett-type plot[54] is in accordance with a photon-limited regime in the case of electron-poor aryl bromides (see Section 2.1.3 Supplementary Information). It is worth noting that the rate-limiting step shifted to oxidative addition in the case of electron-rich aryl bromides, such as 4-bromoanisole.

Based on our experimental findings and the relevant literature, we propose the following mechanistic scenario (Fig. 4E). Under irradiation, reductive quenching of the photocatalyst generates alkyl radicals, which subsequently add to the low-valent nickel species **A**, resulting in the formation of an alkyl-Ni$^I$ species (**B**). Next, **B** undergoes oxidative addition with the aryl bromide, leading to the formation of species **C**, and subsequently **D** through reductive elimination. The reduced photocatalyst then closes the catalytic cycle by converting **D** back to **A** (E(PC/PC$_{red}$) = -1.21 V vs SCE[55]; E(Ni$^I$/Ni$^0$) > -1.1 vs SCE[56]). Further elaboration on this mechanistic sequence is provided in Section 2.1 of the Supplementary Information.

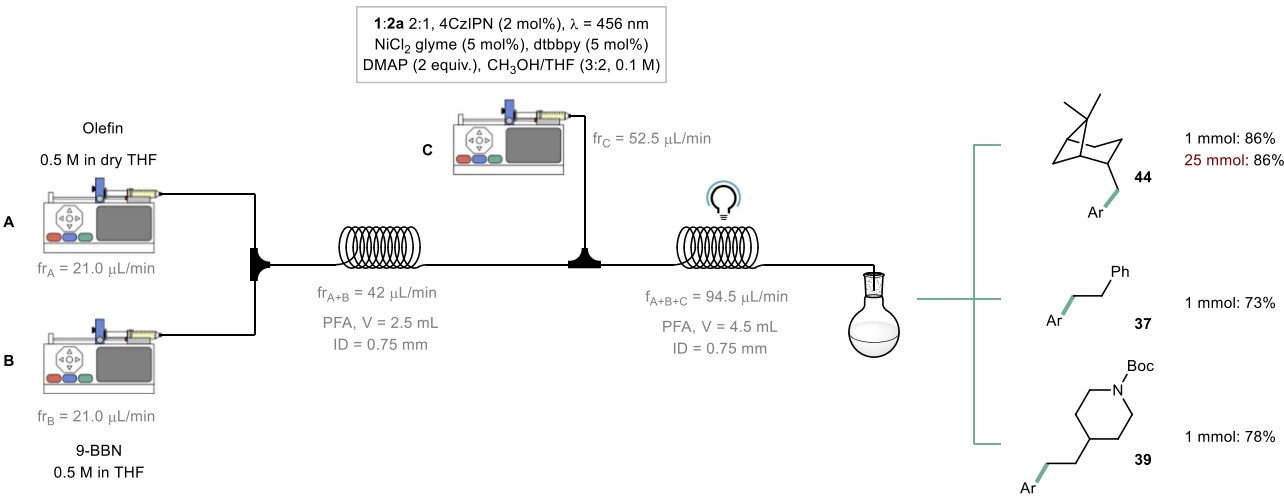

**Fig. 5 | Continuous-flow experiments.** Translation to continuous-flow conditions (see GP5 in the Supplementary Methods) and scale-up (see Section 1.5.5 in the Supplementary Information). Ar: (4-CO$_2$Me)-C$_6$H$_4$. fr flow rate.

## Experiments in continuous flow

Based on the observation of a photon-limited regime in our batch reactions, it becomes apparent that microreactor technology, with its higher photon fluxes, offers significant advantages in terms of rate acceleration. This, in turn, leads to increased throughput and improved scalability of the chemistry[57–59]. Moreover, by integrating the hydroboration step with the metallaphotoredox B-alkyl SMC reaction, the alkyl boranes can be generated on demand and immediately utilized, thereby reducing the handling, purification, and storage requirements for these coupling partners[60–65]. Following a brief optimization of reaction conditions on a 0.1 mmol scale (as described in Section 1.4.3 of the Supplementary Information), we successfully connected the hydroboration and B-alkyl SMC steps. To achieve this, two syringe pumps were employed, each containing a 0.5 M THF solution of β-pinene (Feed A in Fig. 5) and a commercially available 0.5 M THF solution of 9-BBN (Feed B), respectively. The solutions were combined in a PEEK T-mixer, and the resulting reaction mixture was introduced into a 2.5 mL PFA capillary ($t_R$ = 60 min) maintained at an ambient temperature of 22 °C. Subsequently, the stream containing the alkyl borane was merged with another stream containing the necessary reactants for the B-alkyl SMC reaction (Feed C) and introduced into a flow photoreactor ($t_R$ = 48 min, see Supplementary Methods). By employing this streamlined flow process, compound **44** was obtained with an 86% yield after isolation through column chromatography. It is noteworthy that the use of flow conditions allowed for a reduction in both the nickel (5 mol%) and photocatalyst loading (2 mol%), while still yielding comparable results to the batch process. Importantly, this flow approach facilitated easy scalability of the transformation to a 25 mmol scale without the need for reoptimization of the reaction conditions. Finally, we applied this flow approach to other representative scaffolds (**37** and **39**), resulting in good yields ranging from 73% to 78% (Fig. 5). Remarkably, the reaction of **1a** with **2a** was also conducted in flow (see Supplementary Information for experimental details): a residence time of 10 min was enough to obtain product **3** in 86% $^1$H NMR yield.

Overall, we have developed a fast, robust, and scalable photocatalytic B-alkyl Suzuki−Miyaura cross-coupling method that overcomes the limitations of previous approaches, including the requirement for high reaction temperature, expensive catalytic system, and long reaction times. By employing alkyl boranes as alkyl fragments, we have addressed the challenge of arylation at primary C(sp$^3$) centers, expanding the synthetic possibilities of this methodology. The straightforward synthesis of alkyl boranes through

hydroboration makes our approach ideal for functionalizing biologically relevant alkyl substrates in medicinal chemistry and total synthesis. Additionally, we have demonstrated the benefits of continuous-flow technology in terms of scalability, modularity, and reliability. We anticipate that this method for C(sp$^3$)−C(sp$^2$) bond formation is poised to rapidly gain popularity in pharmaceutical route scouting campaigns, offering accelerated drug development opportunities due to its modularity and wide range of available starting materials.

## Methods
### Procedure for Method A
**Step 1: Hydroboration.** 1.1 mL of BH$_3$ solution in THF (1 M, 1.1 mmol) was added to a flame-dried Schlenk flask equipped with a stirring bar at 0 °C (ice bath). The olefin (3.0 mmol) was added neat dropwise. After addition, the ice bath was removed and the reaction was stirred at room temperature for 1 h. In case of solid olefins, the substrate was added to a flame-dried Schlenk flask under a nitrogen atmosphere and dissolved using the minimum amount of anhydrous THF. Hence, THF-BH$_3$ was added dropwise at 0 °C, after which the cold bath was removed. The solution was left stirring for 1 h at room temperature.

**Step 2: Photoreaction.** A CH$_3$CN (0.1 M) solution containing the aryl bromide **2** (0.5 mmol), NiCl$_2$·glyme and dtbbpy (10 mol%), 4CzIPN (5 mol%) and 2,6-lutidine (2 equiv.) was prepared in a 7 mL vial equipped with a screw cap and a stirring bar. The solution was sonicated until fully homogeneous and then bubbled with N$_2$ (5 min). Alkyl borane from Step 1 (in THF) **1** (1 equiv.) was added. Then the solution was irradiated at λ = 456 nm for 3 h. The solvent was removed under reduced pressure and the crude was purified via column chromatography on silica gel to provide the expected product.

### Procedure for Method B
**Step 1: Hydroboration.** 2 mL of 9-BBN solution in THF (0.5 M, 1.0 mmol, 1.0 equiv.) were added to a flame-dried Schlenk flask equipped with a stirring bar at 0 °C (ice bath). The olefin (1.0 mmol, 1.0 equiv.) was added neat dropwise. After addition, the ice bath was removed and the reaction was stirred at room temperature for 1 h. The reaction was monitored via $^1$H-NMR without any purification. In case of solid olefins, the substrate was added to a flame-dried Schlenk flask under a nitrogen atmosphere and dissolved using the minimum amount of anhydrous THF. Hence, 9-BBN was added dropwise at 0 °C, after which the cold bath was removed. The solution was left stirring for 1 h at room temperature.

**Step 2: Photoreaction.** A CH$_3$OH (0.1 M) solution containing the aryl bromide **2** (0.5 mmol), NiCl$_2$·glyme and dtbbpy (10 mol%), 4CzIPN (5 mol%) and 4-dimethylaminopyridine (2 equiv.) was prepared in a 7 mL vial equipped with a screw cap and a stirring bar. The solution sonicated until fully dissolved and then bubbled with N$_2$ (5 min). Alkyl borane from Step 1 (in THF) **1** (2 equiv.) was added. Then the solution was irradiated at $\lambda = 456$ nm for 3 h. The solvent was removed under reduced pressure and the crude was purified via column chromatography on silica gel to provide the expected product.

## Data availability

The authors declare that all the data supporting the findings of this work are available within the article and its Supplementary Information files or from the corresponding author upon request.

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

## Acknowledgements

T.W. has received support from the China Scholarship Council (CSC) for her PhD studies. L.C. acknowledges the framework of the COMP-R Initiatives, funded by the 'Departments of Excellence' programs of the Italian Ministry for University and Research (MUR, 2023–2027). T.N. thank the European Union for funding (GreenDigiPharma, No. 101073089, J.D. and T.N.). We also wish to acknowledge Dr. Dietrich Böse for fruitful discussions (Merck Healthcare KGaA).

## Author contributions

L.C. and T.W. contributed equally in the conceptualization of the project and performed the optimization experiments and the reaction scope. F.J.Z. and B.B. carried out and analyzed the EPR experiments. L.C., J.D. and A.S. performed experiments on the mechanistic investigation and the reaction scope. L.C. and T.N. directed the project and wrote the manuscript with input from all co-authors.

## Competing interests

The authors declare no competing interests.
