## [Peer Review File · Nature Communications]

Rapid and Scalable Photocatalytic C(sp²)–C(sp³) Suzuki
–Miyaura Cross-Coupling of Aryl Bromides with Alkyl BoranesEditorial Note: This manuscript has been previously reviewed at another journal that is not operating a transparent peer review scheme. This document only contains reviewer comments and rebuttal letters for versions considered at Nature Communications.

Reviewers' Comments:

Reviewer #1:

Remarks to the Author:

Noel and co-workers have made a number of additions and clarifications in this revised manuscript which further increased the quality of this paper. The evaluation of additional amine containing substrates is very much appreciated as it gives a more balanced picture on what is possible (with/without N-protection). In addition, the authors have added further details including specific experiments to investigate the underlying mechanisms which seems to be more complex than initially proposed, i.e., radicals appear to be involved as well as various Ni-species. Despite the added complexity, the presentation of this data is effective and this reviewer thinks that there is merit in presenting this more balanced discussion on the mechanism, which as stated by the authors deserves further investigations.

In light of the effective response to the points this reviewer raised, and the high quality of the paper with its various new aspects that will make this methods of relevance for many synthetic chemists, the manuscript can be recommended for acceptance at this stage.

Reviewer #2:

Remarks to the Author:

In this revised manuscript, Noël and coworkers have addressed the large majority of my comments from their initial submission (which I had suggested should be accepted in Nature Communications if these comments were addressed). I appreciate the additional mechanistic insight that the added experiments now provide. I believe the introduction is now more appropriate and more clearly sets the stage for the key contributions of this work. I also agree with the authors' comments (in the response to reviewer comments) that one of the important contributions in this work is the ability to carry out these cross-coupling reactions under milder conditions and in significantly shorter reaction times. I only have a few minor comments remaining that I believe should be addressed prior to publication in Nature Communications.

1. Please provide a reference for the claim that boron derived byproducts have "manageable toxicity concerns".
2. Page 10, line 271 – How do these data suggest that oxidative addition might be the TLS? The reaction conditions being examined are not relevant to the catalytic system at play (i.e., no light or photocatalyst). Couldn't one also imagine that the slow consumption of oxidative addition complex suggests that transmetalation or reductive elimination could be turnover limiting under these thermal/dark conditions?
3. Page 10, line 274 - "this shows that oxidative addition is sluggish" – this is a little misleading as written since (as later shown by the authors), you really need kinetic experiments to make this claim. If the authors want to keep this statement, at a minimum, they should modify it to "oxidative addition under thermal conditions is sluggish".
4. Page 10, line 280 - replace "reiterated" = "iterative"
5. The first half of the conclusion should be reworked to be a little more specific. For example "overcomes the limitations of previous approaches". Please be more specific as to what these were.

Noel and co-workers have made a number of additions and clarifications in this revised manuscript which further increased the quality of this paper. The evaluation of additional amine containing substrates is very much appreciated as it gives a more balanced picture on what is possible (with/without N-protection). In addition, the authors have added further details including specific experiments to investigate the underlying mechanisms which seems to be more complex than initially proposed, i.e., radicals appear to be involved as well as various Ni-species. Despite the added complexity, the presentation of this data is effective and this reviewer thinks that there is merit in presenting this more balanced discussion on the mechanism, which as stated by the authors deserves further investigations.

In light of the effective response to the points this reviewer raised, and the high quality of the paper with its various new aspects that will make this methods of relevance for many synthetic chemists, the manuscript can be recommended for acceptance at this stage.

We thank the Reviewer for their appreciation.

Reviewer #2 (Remarks to the Author):

In this revised manuscript, Noël and coworkers have addressed the large majority of my comments from their initial submission (which I had suggested should be accepted in Nature Communications if these comments were addressed). I appreciate the additional mechanistic insight that the added experiments now provide. I believe the introduction is now more appropriate and more clearly sets the stage for the key contributions of this work. I also agree with the authors' comments (in the response to reviewer comments) that one of the important contributions in this work is the ability to carry out these cross-coupling reactions under milder conditions and in significantly shorter reaction times. I only have a few minor comments remaining that I believe should be addressed prior to publication in Nature Communications.

1. Please provide a reference for the claim that boron derived byproducts have "manageable toxicity concerns".

Done as requested (see ref. 9 in the revised manuscript).

2. Page 10, line 271 – How do these data suggest that oxidative addition might be the TLS? The reaction conditions being examined are not relevant to the catalytic system at play (i.e., no light or photocatalyst). Couldn't one also imagine that the slow consumption of oxidative addition complex suggests that transmetalation or reductive elimination could be turnover limiting under these thermal/dark conditions?

We thank the Reviewer for their comments. Our rationale behind conducting the experiments outlined in Scheme 4C was to gain deeper insights into the operational mechanisms of the nickel catalyst. By eliminating the influence of light, we aimed to elucidate the inherent challenges within the nickel-catalytic cycle. In particular:

- Experiment i) showed that, when starting from stoichiometric Ar-Ni(II)-Br complex, transmetalation (TM) and reductive elimination (RE) can occur in the dark, which suggests these steps occur smoothly already in the dark (stoichiometrically): photocatalysis is not needed to enable these steps.*
- Experiment ii) confirms that the nickel operates in a non-catalytic way in the absence of photocatalysis. Since Experiment i) showed that TM and RE occur smoothly, we suspected that the step inhibiting catalysis was the oxidative addition (OA).*
- Indeed, Experiment iii) showed that a Ni⁰ species (obtained after RE in Experiments i) and ii)) is not able to start catalysis.*
- Experiment iv) shows that photocatalytic conditions can solve this problem related to the OA step by opening an avenue to a radical domain.*

3. Page 10, line 274 - "this shows that oxidative addition is sluggish" – this is a little misleading as written since (as later shown by the authors), you really need kinetic experiments to make this claim. If the authors want to keep this statement, at a minimum, they should modify it to "oxidative addition under thermal conditions is sluggish".

We thank the Reviewer for their suggestion. We made the change, accordingly.

4. Page 10, line 280 - replace "reiterated" = "iterative"

Done as requested.

5. The first half of the conclusion should be reworked to be a little more specific. For example "overcomes the limitations of previous approaches". Please be more specific as to what these were.

We rephrased the paragraph as requested, which now reads as “*We have developed a fast, robust, and scalable photocatalytic B-alkyl Suzuki–Miyaura cross-coupling method that overcomes the limitations of previous approaches, including the requirement for high reaction temperature, expensive catalytic system, and long reaction times.*”